# Technical note: Isotopic fractionation of evaporating waters: effect of sub-daily atmospheric variations and eventual depletion of heavy isotopes.

Francesc Gallart[1], Sebastián González-Fuentes[2], Pilar Llorens[1]

[1]Surface Hydrology and Erosion group, IDAEA, CSIC. Barcelona, 08034, Spain
[2]Groundwater and Hydrogeochemistry group, IDAEA, CSIC. Barcelona, 08034, Spain

*Correspondence to*: Francesc Gallart (francesc.gallart@idaea.csic.es)

**Abstract.** Isotopic fractionation of evaporating waters has been studied constantly in recent decades, particularly because it enables calculation of both the volume of water evaporated from a water body and the isotopic composition of its source water. We studied the stable water isotopic composition of an artificial pan filled with water and subject to total evaporation in a sub-humid environment, in order to put into practice an operational method for estimating the time since disconnection of riverine pools when these are sampled for the quality of aquatic life.

Results indicate that: (i) when about 70% of pan water had evaporated and its isotopic composition became enriched in heavy isotopes, some subsequent periods of depletion instead of enrichment happened, and (ii) the customary application of isotopic fractionation equations to determine the isotopic composition of the water in the pan using weekly averaged atmospheric conditions (temperature and relative humidity) strongly underestimated the changes observed, but predicted an early depletion of heavy isotopes. The first result, rarely reported in the literature, was found to be fully consistent with the early studies of the isotopic composition of evaporating waters. The second one could be attributed to that weekly averages of temperature and relative humidity strongly overestimated air relative humidity during daylight periods of active evaporation. However, when the fractionation equations were parameterized using temperature and relative humidity weighted by potential evapotranspiration at sub-hourly time steps, they adequately reproduced the observed isotopic composition of the water in the pan, including the late periods of heavy isotope depletion.

We demonstrate how weekly increases in air relative humidity when the pan water was already enriched in heavy isotopes led to their depletion. We also analyse the errors that can be incurred if time averages are used instead of flux weighted meteorological data for model parameterization and if unidentified periods of heavy isotope depletion occur. Our results should be taken into account when applying fractionation equations, particularly in conditions or areas with high air relative humidity.

## 1 Introduction

There is increasing interest in the use of stable isotopes in open, soil and xylem waters to study either the volume of evaporated water or the original composition of the source water. The following articles synthesized published studies of open water (Skrzypek *et al*., 2015), of soil and vegetation (Benettin *et al*., 2019) and of rainfall interception processes (Allen *et al*., 2017). Several authors recommended using temperature and relative humidity weighted by the evaporation flux for fractionation studies (e.g. Allison and Leaney, 1982; Gibson, 2002; Gibson *et al*., 2008; 2016). However, in practice this recommendation is not usually followed (except, for example, Mayr *et al*., 2007), particularly where air humidity is low (e.g. Hamilton *et al*., 2005; Skrzypek *et al*., 2015) or when monthly or seasonal periods are investigated (e.g. Benettin *et al*., 2018).

During evaporation of water in a natural environment, lighter water molecules usually vaporize faster than heavier ones, so that the remaining water body becomes enriched in heavier isotopes (Craig *et al.*, 1963). However, heavy isotope concentrations in water evaporating into open air with moderate or high relative humidity increase asymptotically to a stationary isotopic state, establishing a dynamic equilibrium as the mass of water decreases to zero. This effect is attributed to a rapid molecular exchange of isotopes between the water body and the atmospheric vapour, which predominates over the net isotopic effect of a simple evaporation process (Craig *et al.*, 1963). Isotopic exchange that induces the depletion rather than enrichment in heavy isotopes of the evaporating water has been identified, for example, in canopy interception processes, when air humidity is near to saturation (Saxena, 1968; Allen *et al.*, 2017).

Gonfiantini (1986) showed that stationary or limiting isotopic composition ($\delta^*$) depend mainly on air relative humidity to the extent that, when air relative humidity is low, $\delta^*$ concentrations are practically unreachable; whereas, when it approaches 95%, these can be reached when only about 20% of water had evaporated.

Most temporary rivers, from the cessation of flow to the total desiccation of the river bed, undergo a disconnected pools phase (Gallart *et al.*, 2017). As the aquatic life in these pools naturally changes after the flow cessation (Bonada *et al.*, 2020), the time between disconnection of these pools and sampling is information needed for assessing their biological quality. To implement an operational procedure to estimate the time since disconnection of river pools, based on the study of water stable isotopes, an artificial pan was set up and subjected to evaporation to complete dryness in a location with a sub-humid climate, as a counterweight to the more frequent studies in dry climates.

The purpose of this technical note is to describe and analyse the occurrence of periods of heavy isotope depletion instead of enrichment during evaporation of water in an artificial pan, and the validation of the Gonfiantini (1986) equation in predicting both the enrichment and depletion in heavy isotopes of evaporating waters. Besides, we describe the environmental conditions that induced depletion instead of enrichment in heavy isotopes of the evaporating water and we also analyse the likely errors that can be incurred in studies of the isotopic composition of evaporating water if the meteorological data are time-averaged instead of flux-weighted, as well as when periods of heavy isotope depletion occur during water evaporation.

## 2 Data and Methods.

The experiment was performed in the Vallcebre Research Catchments (Eastern Pyrenees, Iberian Peninsula). Climate is sub-humid Mediterranean; mean temperature is 9.1ºC, mean annual precipitation is 880 mm and mean annual potential evapotranspiration is 818 mm (Llorens *et al.*, 2018).

A round steel pan (Figure 1) was filled with 70 L of water from a nearby spring, partly buried in the ground and protected against precipitation by a lid of clear methacrylate installed 0.5 m above the pan's surface, to allow air circulation. A net stopped animals drinking. Bulk rainfall was sampled with a 180-mm diameter funnel connected to a 1-L plastic bottle with a pipe with a loop to prevent evaporation.

Every week from June to October 2020, until the pan had practically dried out (19 weeks), the volume of water in the pan was recorded and water samples were taken from the pan and bulk rain for isotopic analyses.

Air temperature (T) and relative humidity (h) (Vaisala, Finland), net radiation (Kipp and Zonen, The Netherlands) and wind speed (Thies Clima, Germany) were measured every 10 s and recorded at 5-min intervals by a data logger (Data Taker, Australia) in a meteorological station adjacent to the sampled pan.

Water samples were analysed for their stable isotope ratios ([18]O and [2]H) via cavity ring-down spectroscopy (Picarro L2120-i, Picarro Inc., USA) at the laboratory of the Centre of Hydrogeology, University of Málaga. The precision of the isotope ratio measurements was reported as < 0.1‰ for $\delta^{18}O$ and < 0.4‰ for $\delta^2H$. The data were expressed in the $\delta$-notation as parts per mil (‰) relative to Vienna Standard Mean Ocean Water (VSMOW).

The isotopic composition ($\delta_L$, ‰) of the residual water in the pan was modelled weekly for $^{18}$O and $^2$H, as in Gonfiantini (1986):

$$\delta_L = (\delta_0 - \delta^*)(1-x)^m + \delta^* \tag{1}$$

where $\delta_0$ is the initial or modelled isotopic composition for the end of the previous week, $\delta^*$ the stationary or limiting isotopic composition reached when the remaining water in a pool tends to 0, $m$ is the slope of the temporal enrichment (Gibson *et al.*, 2016), and $x$ is the fraction of water volume that evaporated. Thus, 1-$x$ is the residual volume fraction. It is worth emphasizing that this equation does not require that evaporation induces an increase of $\delta_L$ with respect to $\delta_0$ (enrichment) but that $\delta_L$ approaches $\delta^*$ in either direction (enrichment or depletion).

The coefficients of equation (1) were calculated as follows:

$$\delta^* = \frac{\left(h\delta_A + \varepsilon_k + \varepsilon^+/\alpha^+\right)}{\left(h - 10^{-3}\cdot\left(\varepsilon_k + \varepsilon^+/\alpha^+\right)\right)} \tag{2}$$

$$m = \frac{\left(h - 10^{-3}\cdot\left(\varepsilon_k + \varepsilon^+/\alpha^+\right)\right)}{(1 - h + 10^{-3}\cdot\varepsilon_k)} \tag{3}$$

Where $h$ (-) is the relative humidity, $\delta_A$ (‰) is the isotopic composition of the atmospheric moisture, $\varepsilon_k$ (‰) the kinetic fractionation factor, $\varepsilon^+$ (‰) the isotopic separation between liquid and vapour and $\alpha^+$ (‰) the equilibrium fractionation factor.

$\delta_A$ was assumed to be in equilibrium with the isotopic composition of precipitation $\delta_P$ and was calculated as in Gibson *et al.* (2008):

$$\delta_A = (\delta_P - \varepsilon^+)/\alpha^+ \tag{4}$$

$\varepsilon_k$ was calculated as in Gat (1996) and Horita *et al.* (2008):

$$\varepsilon_k = \theta n (1-h)(1 - D_i/D)10^3 \tag{5}$$

Where $\theta$ is a weighting term that can be assumed equal to 1 for a small body of water whose evaporation flux does not perturb the ambient moisture significantly (Gat, 1996). The factor $n$, which accounts for the aerodynamic regime above the evaporating liquid–vapour interface, was assumed to be 0.5 (turbulent) as for an open water body. $D_i/D$ is the diffusivity ratio between light and heavy isotopes. Following (Merlivat, 1978) $D_i/D$ is 0.9755 and 0.9723 for $^2$H and $^{18}$O, respectively.

$\varepsilon^+$ was obtained from:

$$\varepsilon^+ = (\alpha^+ - 1)10^3 \tag{6}$$

where $\alpha^+$ is calculated from the absolute temperature T (°K), as in Horita & Wesolowski (1994):

$$10^3\ln(\alpha^+) = 1158.8\left(T^3/10^9\right) - 1620.1\left(T^2/10^6\right) + 794.84\left(T/10^3\right) - 161.04 + 2.9992\left(10^9/T^3\right) \tag{7}$$

for $^2$H, and

$$10^3\ln(\alpha^+) = -7.685 + 6.7123\left(10^3/T\right) - 1.6664\left(10^6/T^2\right) + 0.3504\left(10^9/T^3\right) \tag{8}$$

for $^{18}$O.

As we observed some significant periods of heavy isotope depletion in the pan water, we analysed the light isotope balance to investigate the net mass fluxes, i.e to verify whether these isotopes had evaporated rather than condensed throughout the event. For this purpose, the mass of water in moles $M_w$ was obtained for every visit from its volume using a density of 0.9976 kilograms per litre and a molar mass of 18.015 grams per mole. Small changes in these values due to the variation in heavy isotope concentrations were not taken into account because they are mutually cancelled out.

Then, $R_{sa}$ sample isotope ratios where obtained for each of the two heavy isotopes from the corresponding $\delta$ values:

$$R_{sa} = (\partial/1000 + 1) \cdot R_{st} \tag{9}$$

Where $R_{st}$ are the isotopic ratios of the VSMOW standards, which were taken as 1/6420 for $^2$H and 1/498.7 for $^{18}$O. Finally, the mass in moles of the light isotopes $M_l$ were obtained for every sample and isotope using:

$$M_l = M_w \cdot na / (1 + R_{sa}) \tag{10}$$

Where *na* is the number of atoms in each water molecule: 1 for oxygen and 2 for hydrogen.

As time-averaged  meteorological variables were largely biased due to the diel variations (lower temperature and higher relative humidity during the night when evaporation is the lowest), flux-weighted daily and weekly T and h values were obtained by weighting sub-hourly readings with the corresponding evaporative flux, calculated using the Penman-Monteith equation parameterized as for reference evapotranspiration (Allen *et al.*, 1998). A pan coefficient was not necessary because it cancelled out during weighting. Subsequently, the results obtained with time-averaged meteorological variables are labelled "unweighted" and those obtained by PET weighting are labelled "weighted" or unlabelled.

We first applied the Gonfiantini (1986) equation (1) to simulate pan water isotopic composition from observed volume changes, using unweighted and weighted meteorological conditions, allowing analysis of errors in both parameterizations. Furthermore, we analysed the errors on the inverse calculation of the volume of evaporated water from its isotopic composition, which is the most frequent target as stated in the introduction. For this purpose, we applied an inversion of the equation (1):

$$(1 - x) = \left( \frac{\partial_L - \partial^*}{\partial_O - \partial^*} \right)^{(1/m)} \tag{11}$$

Where $\delta_O$ represents the isotopic composition of the pool water at the beginning of the experiment, i.e. the 'original' water in usual applications that can be obtained by the intersection of the Local Evaporation Line (LEL) and the Local Meteoric Water Line (LMWL) (e.g. Benettin et al., 2018), and $\delta_L$ represents the isotopic composition of the water obtained at every sampling visit. The other variables and parameters are the same as for equation (1).

## 3 Results and Discussion.

The blue line in Figure 2a shows the chronicle of the isotopic composition ($\delta^{18}$O) of the water sampled in the pan. During the first weeks the water became rapidly enriched in $^{18}$O but after about two and a half months (71 days), when the relative volume of water in the pan was below 20%, it underwent alternating periods of enrichment and depletion. At the end of the experiment, the isotopic composition of the pan water was similar to that observed 10 weeks earlier. Figure A1 shows that both light isotopes evaporated rather than condensed during the experiment, including late periods of heavy isotope depletion.

Figure 2a shows that the limiting compositions $\delta^*$ obtained with equation (2) using unweighted T and *h* conditions were higher than the compositions observed for the pan water during the first 8 weeks, but were lower during the rest of the experiment. The pan water isotopic compositions were adequately predicted during the first 5-6 weeks, but were clearly underestimated thereafter when the depletion periods were exaggerated.

Figure 2b demonstrates that both the observed and simulated isotopic compositions of the pan water were located near the same LEL, regardless of whether the water became enriched or depleted.

Given the high dependence of the $\delta^*$ limiting values on air relative humidity (Figure 3), an incorrect assessment of the effective value of *h* was deemed the most likely cause of its underestimation using equation (2) and the propagation of the error to the pan water  isotopic composition using equation (1).

The difference between time-averaged and flux-weighted relative humidity is shown for September 4, a late-summer day, in Figure 4. For this day, average daily *h* was 74%, whereas when *h* was weighted with potential evaporation the mean value was 52%. With equation (2) these h values correspond to $\delta^{*18}$O concentrations of 5.94 and 22.15 ‰, respectively. Figures A2 and A3 show, for all the sampling dates, the differences in the daily values of T and *h* when obtained by either time-averaging or evaporation flux-weighting and the corresponding differences in the estimates of $\delta^*$ concentrations.

Temperatures became similarly higher when flux-averaged, regardless of their magnitude. In a different way, relative humidity was habitually lower when flux-averaged, but did not change when it was very high or very low.

Subsequently, following the recommendation made by some authors (e.g. Allison and Leaney, 1982; Gibson, 2002; Gibson *et al*., 2008), we used T and *h* weighted with potential evaporation for parameterizing equation (1). Differences between weighted and unweighted daily values of $\delta^*$ were independent on the daily amplitudes of *h* (Figure A4a), but were null when *h* was close to saturation and significantly increased as *h* decreased (*F*igure A4b).

Figure 5a (chronicles) and 5b (dual plot) show the series of $\delta^{18}O$ observed and simulated and $\delta^*$ for the studied period,
showing that results clearly improved here where equation (1) was applied with T and h weighted, to the point of adequate simulation. Differences between unweighted and weighted $\delta^*$ values were not propagated to those of $\delta_L$ (r=0.04) due to large differences in initial $\delta_0$ values. Figure A5 compares simulated *versus* observed compositions for both isotopes and parameterizations; as commented before, the errors associated to the unweighted T and *h* parameterization were evident after the eighth week. When the weighted parameterization was applied, mean absolute errors decreased by a quotient of 4, from
19.31 and 3.78‰ for $\delta^2H$ and $\delta^{18}O$, respectively, to 4.88 and 0.87‰.

In order to understand the environmental conditions that led to the depletion of heavy isotopes during evaporation, we analysed the dependence of the limiting $\delta^*$ values on their main drivers *h* and $\delta_A$ along the experiment. Figure 6 shows that during the experiment *h* ranged between 0.45 and 0.74 and $\delta_A$ between -20.5 and -13.6, whereas $\delta^*$ had a much larger range between 3.34 and 24.7. This graph demonstrates that the large variability in $\delta^*$ during the experiment shown in Figure 4 was
essentially originated by the variations in *h*, whereas variations in $\delta_A$ had a much smaller influence ($R^2$=0.25) although significant at the 5% level (p=0.035).

Additionally, Figure 7, a diagram designed by A. Rodhe for analysing the isotopic fractionation in forest throughfall (in Saxena,1986), summarizes the role of *h*, $\delta_A$, $\delta^*$ and the isotopic composition of the pan water on the $^{18}O$ enrichment and depletion events during the experiment. According to equation (1), evaporation will approximate the isotopic compositions
of the pan water towards $\delta^*$, by enriching point values that lie below the $\delta^*$ line and depleting those above it. From bottom to top, at the beginning of the experiment, the isotopic composition of the pan water was so depleted that it became enriched in spite of high *h* conditions and the lowest limiting $\delta^*$ values. After the tenth week, the pan water was so enriched that successive large variations in h forced alternated periods of enrichment and depletion. This analysis supports the validity of this graph for qualitatively predicting whether evaporation will determine isotope depletion or enrichment during given
conditions, as proposed by Saxena (1986),

Finally, the errors than can be committed using equations (1) and (2) when parametrised with time-averaged T and h for simulating the relative volumes of water evaporated from the isotopic change are shown in Figure 8. When time-averaged meteorological data were used, the equation (11) predicted slightly smaller residual relative water volumes than observed for the 7 first observations, but it was in mathematical error for the remaining observations. These errors are due to the fact that
the limiting water isotopic composition $\delta^*$ had values between the original $\delta_O$ and the terminal $\delta_L$ ones, an impossible arrangement because, following equation (1), evaporation will approximate the isotopic composition of water towards $\delta^*$ by increasing or decreasing its value, but it cannot modify the isotopic composition of water by crossing the $\delta^*$ value. Figure 2a shows indeed that (unweighted) $\delta^*$ values were smaller than the observed $\delta$ values for the latter two-thirds of the experiment. When the equation (11) was applied with PET-weighted meteorological data, the results were much better for most of the
observations, although some mathematical errors also occurred (line discontinuities). These errors do occur because the actual $\delta_O$ of the evaporating process at every step is not the value at the start of the experiment, but the $\delta_L$ value of the former step. In Figure 5a, these samples correspond to those that underwent $^{18}O$ depletion instead of enrichment: the corresponding observed $\delta$ values where smaller than the preceding ones but larger than the corresponding limiting $\delta^*$ values, so the application of equation (11) yields adequate results at the weekly step scale (depletion). The analysis at the weekly step scale

is not feasible when time-averaged (unweighted) meteorological data were used, because the limiting $\delta^*$ values were strongly underestimated.

In fact, Figures 5a and 8 recall the limitations of the isotopic method for assessing the residual water volume in a pool from its isotopic composition or vice versa. This method would require a monotonic change in the pool water isotopic composition for decreasing water volume but, as already shown by Gonfiantini (1968), this requirement fails if the pool

water isotopic composition approaches the limiting $\delta^*$ value due to a low residual volume associated with high relative air humidity. When this occurs, evaporation may continue with enrichment, stability or depletion in heavy isotopes (Figure 7).

4 Final Remarks.

Current knowledge establishes that water that evaporates in the open air tends to reach limiting or stationary heavy isotopic

compositions ($\delta^*$), which approach those of precipitation ($\delta_P$) when it is in equilibrium with air humidity near to saturation. Under drier conditions, these $\delta^*$ values increase rapidly with decreasing air humidity and become poorly sensitive to atmospheric moisture isotopic composition.

Evaporation of water does not always induce heavy isotope enrichment, but may progress without isotopic change in a steady state process when the composition of evaporating water is equal to the limiting $\delta^*$ value. When this limiting value is

215 exceeded, back-equilibration overwhelms the evaporative distillation and the evaporating water becomes depleted.

In this experiment, we observed several alternating weeks of heavy isotope enrichment and depletion during evaporation of pan water. These events were successfully simulated using classical equations and attributed to temporal increases of air relative humidity and corresponding decreases of the limiting $\delta^*$ values, below the composition reached by the evaporating water.

It was also possible to make qualitative graphical predictions of enrichment or depletion in heavy isotopes during periods of evaporation without the need to know the volumes of water involved.

The success of the isotopic fractionation equations needed the use of flux-weighted temperature and relative humidity conditions during the weekly periods of the experiment. When time-averaged meteorological conditions were used, the errors in the simulation of water isotopic composition were small at the beginning of the experiment when pan water was

225 rather depleted, but became very significant when more of half the volume of water had evaporated; this may be attributed to both the cumulative effect of underestimated limiting compositions on the already underestimated pan water isotopic composition and more strongly propagated errors when water volumes are low.

When using the isotopic composition of evaporating waters to calculate the volume of water evaporated, the susceptibility to periods of heavy isotope depletion must be taken into account. Fortunately, when the target is to calculate the isotopic

composition of the source waters, these depletion periods follow the same LEL slope as enrichment periods.

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

*Author contributions*. PL and FG designed and carried out the field experiment. SG, FG and PL did the calculations and model implementation. FG prepared the initial draft of the article. All the authors revised and accepted the final document.

*Competing interests*. The authors have the following competing interests: at least one of the (co-) authors is a member of the editorial board of Hydrology and Earth System Sciences.

*Acknowledgements*. We are grateful to Gisel Bertran, Jérôme Latron, and both the TRivers-P project and FEHM teams for their assistance. We are also grateful to Michael Eaude for his English style improvements.

*Financial support*. This research was supported by the TRivers-P project (ACA210/18/00022), funded by the Catalan Water Agency (ACA), and by the Rhysotto project (PID2019-106583RB-100) funded by the Spanish Agencia Estatal de Investigación.

**Figures**

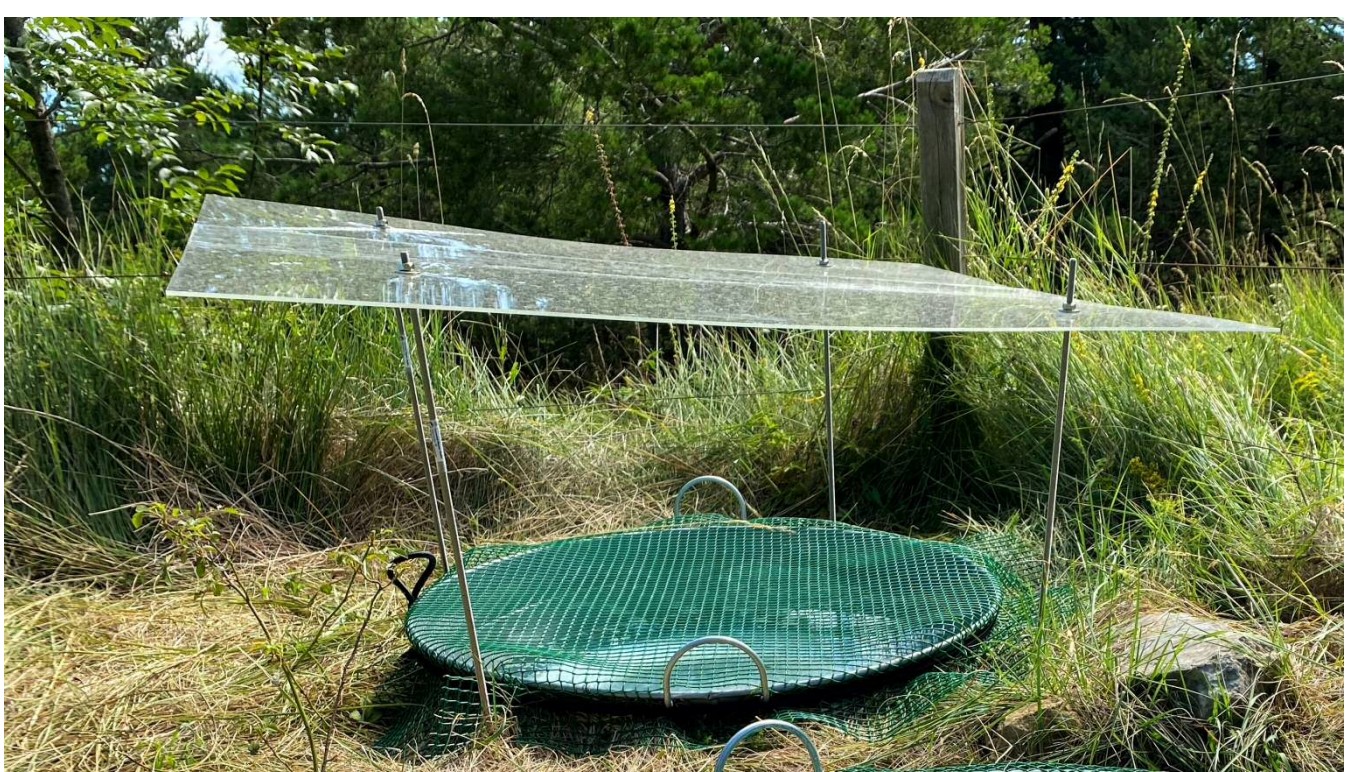

**Figure 1: View of steel experimental pan. The pan was partly buried in the ground and protected against precipitation by a lid of clear methacrylate installed 0.5 m above the pan surface, to allow air circulation, and against large animals drinking by a net.**

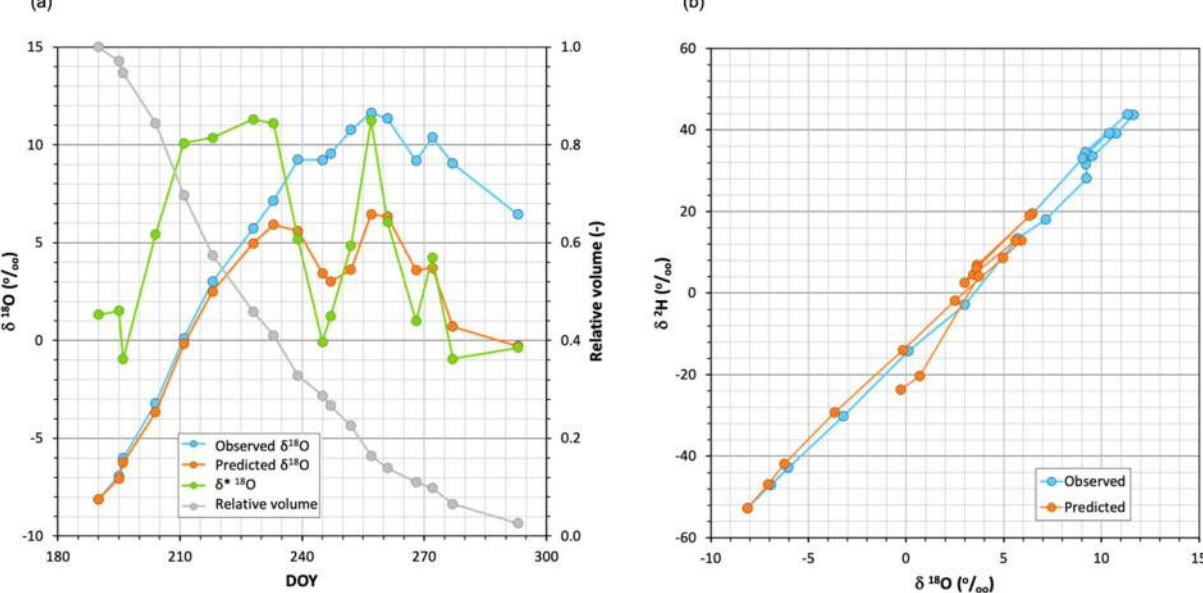

**Figure 2: a) Time series of $\delta^{18}O$ observed in the pan water together with (unweighted) predicted and limiting $\delta^{*}$ $^{18}O$ values when equation (1) was applied, using the weekly-averaged air temperature and relative humidity of the period studied. The relative residual volume curve is also indicated. b) Dual plot of the $\delta^{18}O$ observed and predicted during the period studied.**

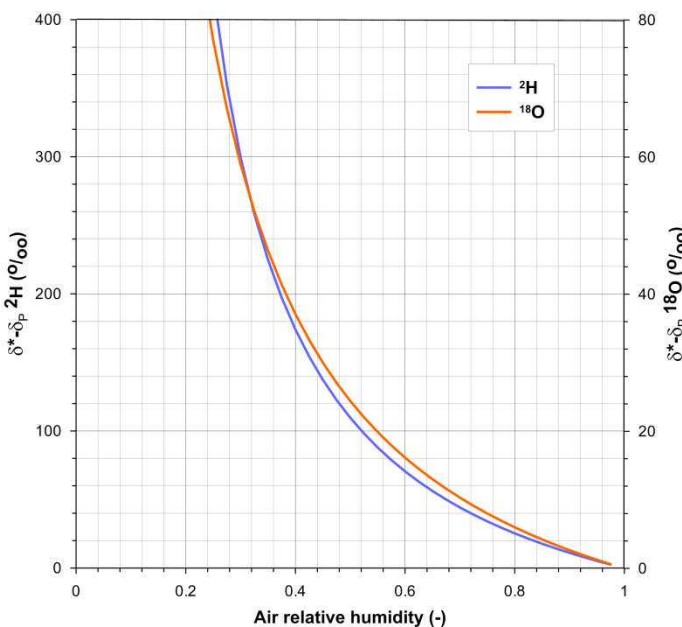

**Figure 3: Relationship between air relative humidity and the differences between the limiting isotopic compositions ($\delta^{*}$) and the isotopic composition of precipitation ($\delta_{P}$), for $^{2}H$ and $^{18}O$ calculated using equation (2). Temperature is set at 20°C.**

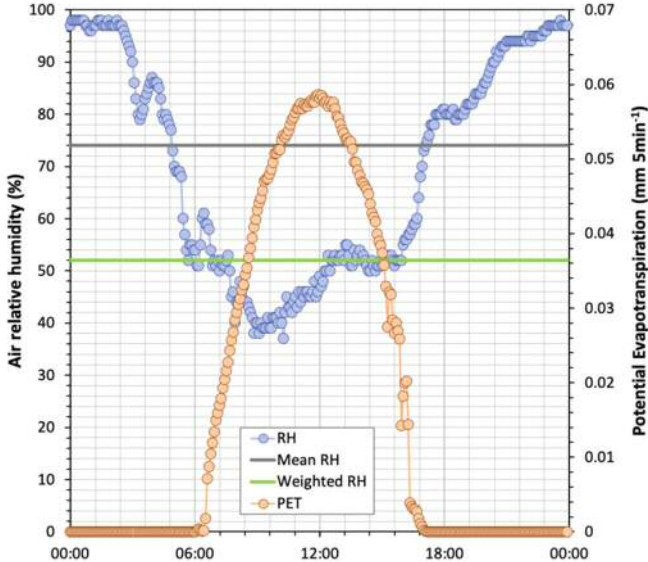

**Figure 4: Time series of air relative humidity and potential evaporation for 4 September, 2020 (five-minute time steps). The mean relative humidity and relative humidity weighted by evaporation flux are also shown.**

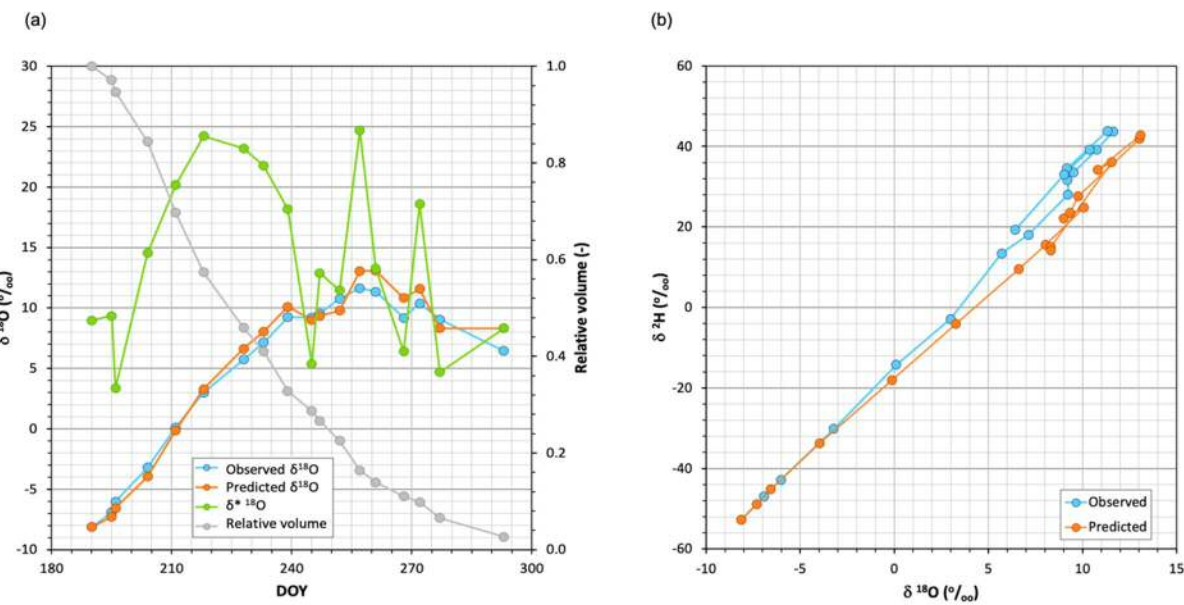

**Figure 5: a) Time series of $\delta^{18}O$ observed in the pan water together with (weighted) simulated and limiting $\delta^* \, ^{18}O$ values when equation (1) was applied, using evaporation flux-averaged air temperature and relative humidity conditions. b) Dual plot of the $\delta^{18}O$ observed and predicted during the period studied.**

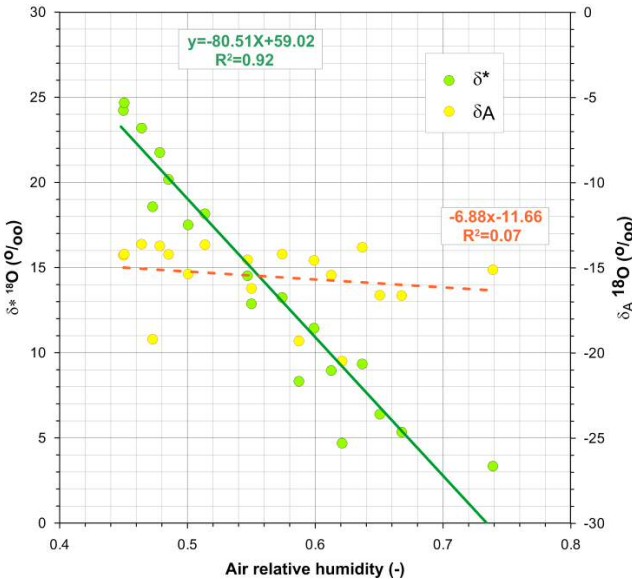

**Figure 6: Weekly estimates of <sup>18</sup>O isotopic composition, limiting (δ\*) and in air moisture (δ<sub>A</sub>), in relation to relative humidity. Vertical scales are offset 30 <sup>0</sup>/<sub>00</sub> from each other.**

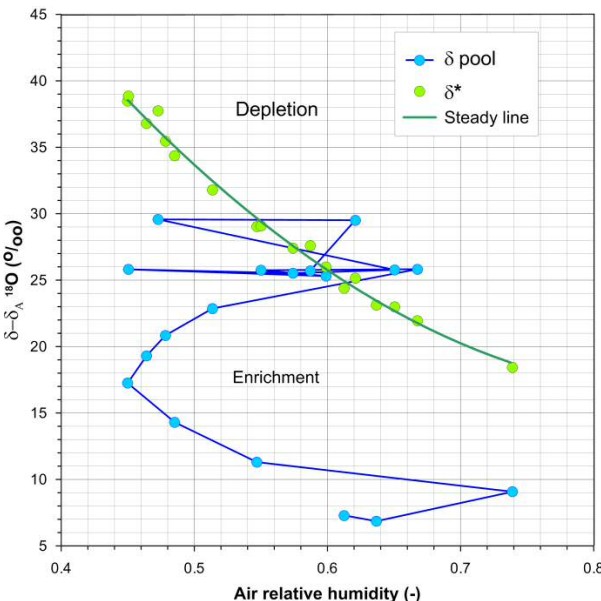

**Figure 7: Evolution of the weekly differences between the isotopic composition of water and air, plotted in relation to air relative humidity (from bottom to top). The green line that connects the δ\*-δ<sub>A</sub> points is a second order polynomial, shown only as visual reference.**

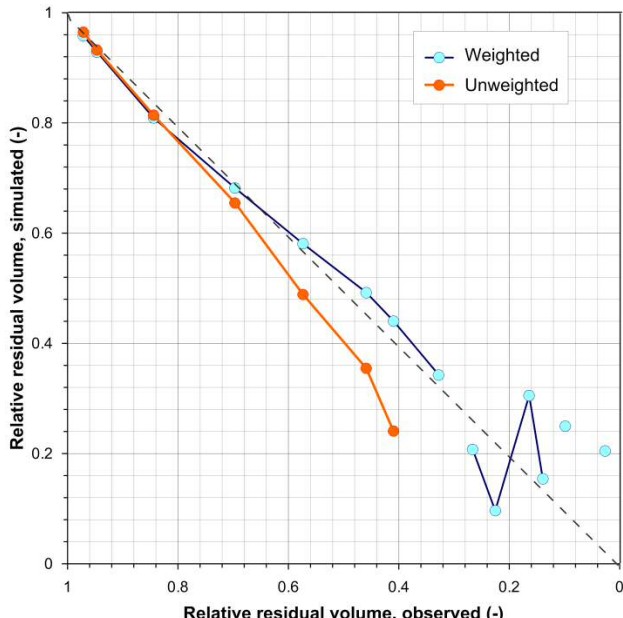

**Figure 8: Simulated versus observed residual volumes of evaporating water applying equation (11) and comparing the use of both unweighted and PET-weighted meteorological parameters. Gaps in the lines correspond to lacking points due to mathematical errors (non-real number results) of equation (11).**

## Appendix A

Some complementary figures included in this Appendix give further details on the experiment.

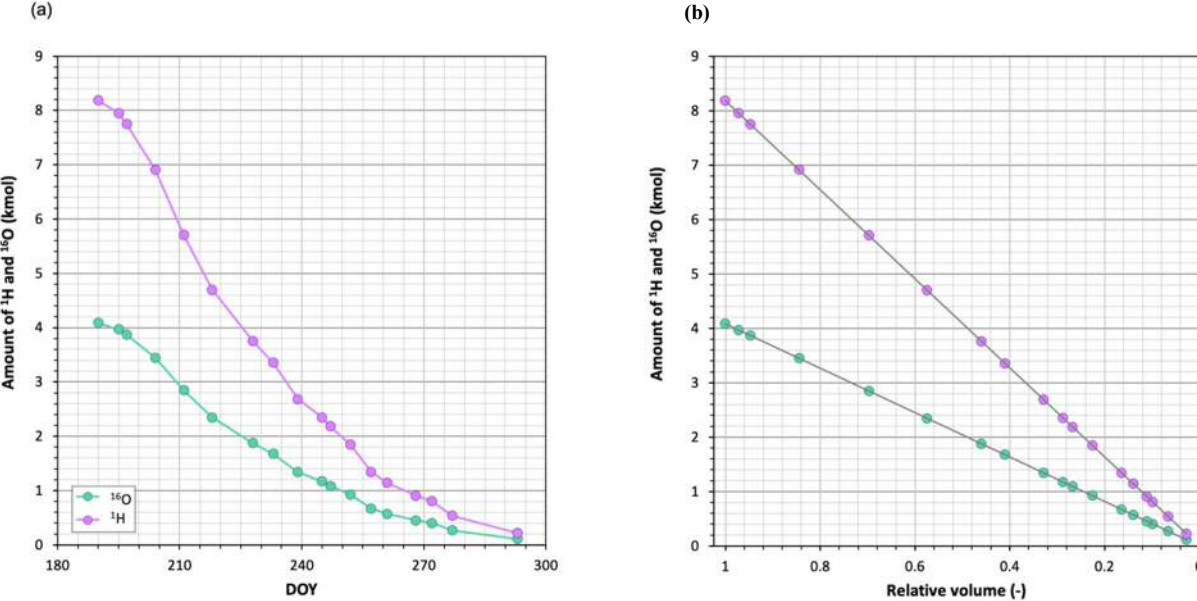

**Figure A1: Amount of light isotopes in the water during the experiment, a) for the date and b) for the relative residual volume, calculated using Equation (10). Evaporation rates always exceeded condensation ones, even during the late periods of heavy isotope depletion.**

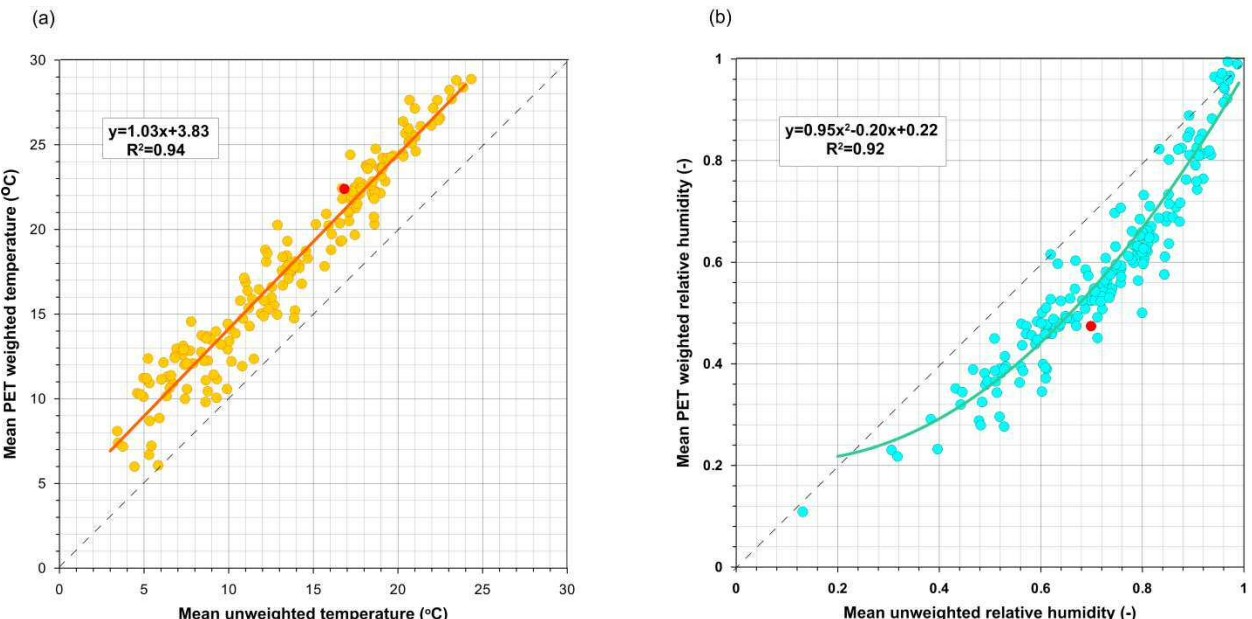

**Figure A2: Comparison of the differences in daily a) temperature and b) relative humidity when 5-minute readings were time-averaged (unweighted) or weighted with potential evapotranspiration. The red dots represent the conditions on the date shown in Figure 5. The dashed line shows the 1:1 relationship.**

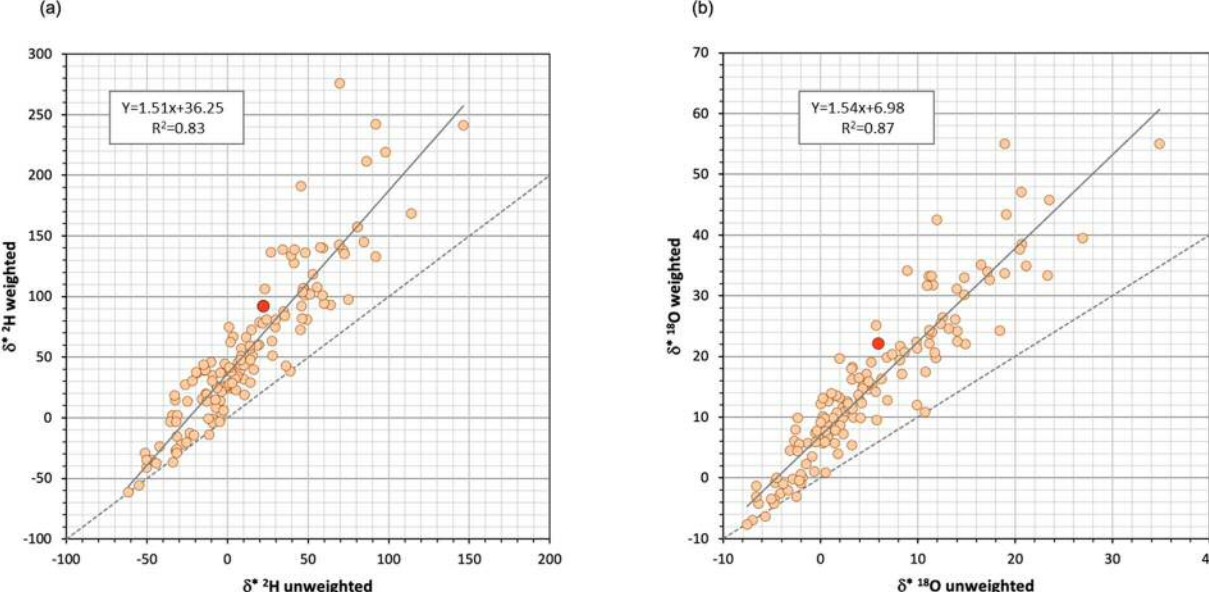

**Figure A3: Comparison of the daily limiting compositions δ\* for a) ²H and b) ¹⁸O when 5-minute readings were time-averaged (unweighted) or weighted with potential evapotranspiration. Low δ\* values show the least differences because these correspond to days with high relative humidity, as shown in Figure 3 and A4. The red dots represent the conditions on the date shown in Figure 5. The dashed line shows the 1:1 relationship.**

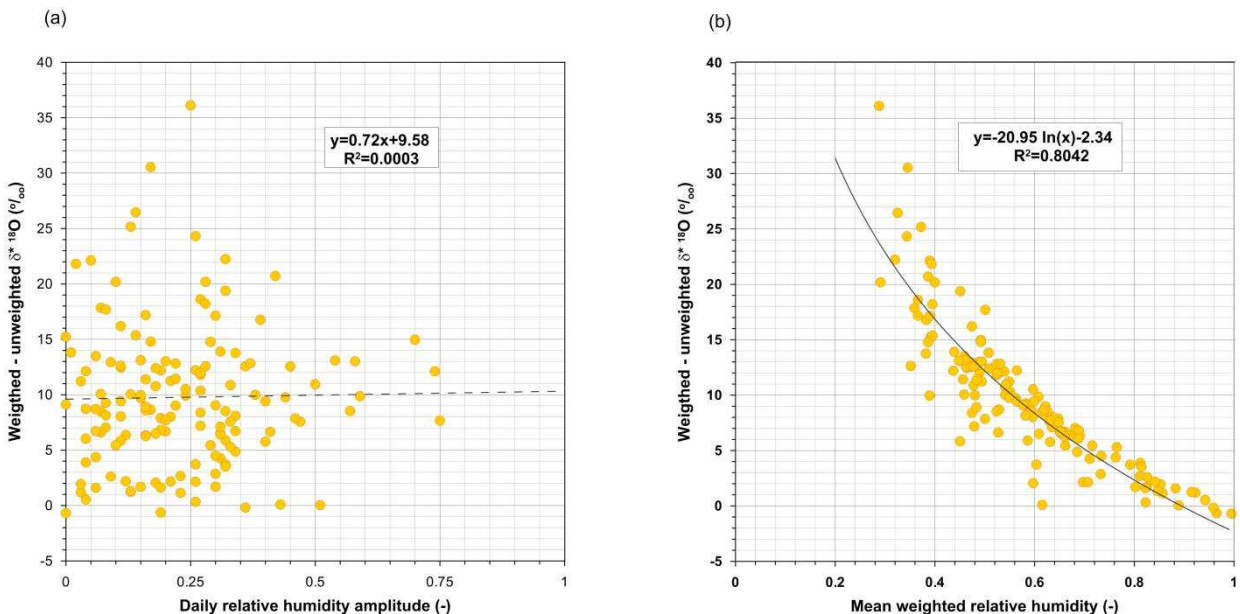

**Figure A4: Dependence of the differences between the daily limiting δ\* values obtained with-time averaged meteorological conditions (unweighted) and those weighted by PET (weighted) on a) the daily amplitudes of relative humidity and b) the weighted daily relative humidity (b). Equations are shown only to demonstrate the dependences.**

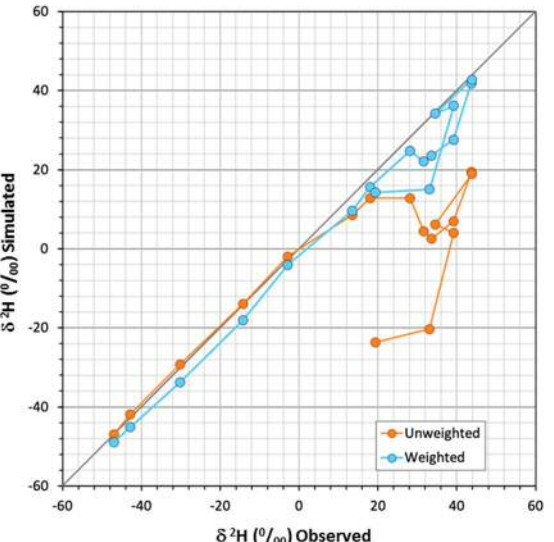 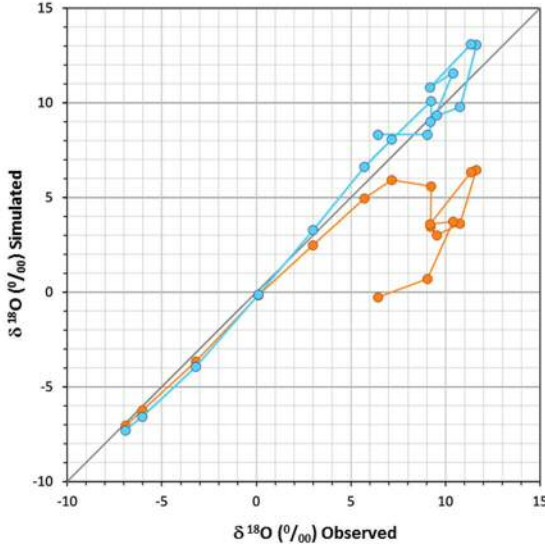

**Figure A5: Simulated *versus* observed heavy isotope concentrations when air temperature and relative humidity are time-averaged or weighted with potential evapotranspiration.**