# Peer review of "Technical note: Isotopic fractionation of evaporating waters: effect of sub-daily atmospheric variations and eventual depletion of heavy isotopes."

_Hydrology and Earth System Sciences, 2022_

## Author Comment (AC1)

We thank the reviewer for his comments, which show us some aspects of our note that we did not explain well enough.

- The intended more original aspect of the note is not the role of diurnal variations, but rather the evidence of periods of heavy isotope depletion during water evaporation, along with the adequate prediction of these periods using customary fractionation equations parameterized with air conditions.

In the recent literature, heavy isotope depletion of water is sometimes attributed to exchange with atmospheric moisture in equilibrium conditions, whereas our results show that it can happen during evaporation when relative humidity is still far from saturation. The early definitions of $\delta*$ concentrations are not only the "isotopic composition which a water reaches in its final evaporation stages", but also "If h> 50% the isotopic composition rapidly approaches the limiting value A/B ($\delta*$) and then remains practically constant, simulating the occurrence of an *isotopic steady state* in the final evaporation stage" (Gonfiantini, 1986).

These concepts are very relevant, because If the $\delta*$ concentrations are understood as only reached in the "final evaporation stages" (as reiterated in several recent articles) there is no way to understand how evaporation can result in depletion of heavy isotopes. Conversely, if these $\delta*$ concentrations are understood as "an isotopic steady state" or dynamic equilibrium that can be rapidly approached when the relative humidity of the air is rather high, It is easy to understand that changes in atmospheric conditions can induce the decrease of $\delta*$ concentrations below the concentrations already reached in the pan water, and determine the depletion of heavy isotopes during evaporation.

Several authors recommended the use of evaporation flux-weighted atmospheric conditions for isotopic fractionation studies, but we found no evaluation of the errors that can be incurred if these recommendations are not followed.

-The sentence "rapid molecular exchange of isotopes between the water body and the atmospheric vapour, which predominates over the net isotopic effect of a simple evaporation process" is not ours, but quoted from Craig et al., 1963. We acknowledge the proposal of more updated citations, but we try to show that our findings can be explained by the early developments made many years ago on the isotopy of evaporating open water bodies.

-We did not explain well ourselves with our sentence "In drier conditions, these $\delta*$ rapidly increase with decreasing air humidity and become detached from precipitation and atmospheric moisture isotopic content". We wanted to remark that, following eq. (2) and the resulting Fig (3), as the air becomes drier, the $\delta*$ values sharply increase to an extent that the variations of atmospheric moisture isotopy become poorly relevant. Indeed, the results in our experiment show that the observed temporal changes in air relative humidity become much more important than those in air moisture isotopy, as estimated from precipitation isotopy. Figure R 1 shows the dependence of weekly $\delta*^{18}O$ and air moisture $\delta^{18}O$ estimates on relative

humidity along the experiment. Air relative humidity explains 92% of $\delta^*$ $^{18}$O variance while moisture isotopy explains only 25% (not shown in this graph).

[Figure]

**Figure R 1: weekly estimates of $^{18}$O concentrations; steady ($\delta^*$) and in air moisture ($\delta a$), in relation to relative humidity. Vertical scales are offset 30 $^0/_{00}$ from each other.**

- We thought that showing the temporal variation of $\delta^*$ values was sufficient to explain why we have decreasing isotopic composition at the late intervals. The reviewer's comment leads us to add a new Figure R 2, similar to the one proposed by A. Rhode (cited by Saxena, 1986), which can help to understand the behaviour of our pan water.

[Figure]

**Figure R 2: Evolution of the weekly differences between the moisture isotopy of water and air, plotted in relation to relative humidity (from bottom to top).**

In this figure, the differences in $\delta$ $^{18}$O between water and air moisture are plotted against air relative humidity. The differences between $\delta^*$ and air moisture $\delta$ values split the plot into a lower part where the differences in pan water isotopy are lower than that of $\delta^*$ and an upper part where these are higher. According to equation (1), water conditions located on the $\delta^*$

curve undergo evaporation without any isotopic change (steady state or dynamic equilibrium), while evaporating waters located outside this curve tend to move towards it, becoming enriched (upward) or depleted (downward). Minor vertical displacements of the points may be due to changes in the isotopy of the air moisture.

The isotopically depleted initial conditions (lower points) of the pan water determined its progressive enrichment during evaporation in spite of wide changes in air relative humidity. But by the ninth week of the experiment, pan water was already so enriched that a relevant increase in air relative humidity caused a strong decrease of $\delta^*$ and moved the situation of the pan water point to the 'depletion' area of the graph. Since this event, successive alternations in relative humidity determined isotopic enrichment and depletion periods, as already shown in Figures 2 and 6. We hope that these two new figures will help to show the changing environmental conditions along the experiment.

- We used air relative humidity because this is requested in the equations we used. Before using reference evapotranspiration for weighting relative humidity we tested several simpler options such as global radiation, vapour pressure deficit, as well as the combination of mean, minimum and maximum daily relative humidity. We agree that Penman formulation would be more adequate than Penman-Monteith reference formulation, but parameterisation of Penman-Monteith equation to 5-minute steps is more physically sound and, although we did not use it for estimating pan evaporation, it predicted the decrease of its level with a determination coefficient of $r^2$=0.994.

- Pan level and the isotopy of both pan and precipitation waters were measured at weekly intervals, so it was necessary to aggregate air temperature and relative humidity data to these periods for applying equations. The values of $\delta^*$ were obtained for every weekly step using equation (2). Pan evaporation $x$ was obtained from observed water level changes in the pan and the simulated isotopy $\delta_L$ of pan water was calculated using equation (1). For the first step, the original isotopic composition of the pan water was used as the initial isotopy $\delta_0$, and for the subsequent time steps, the isotopy $\delta_L$ obtained in the former step was used as the initial isotopy $\delta_0$ for the new step.

We hope these explanations will help readers better understand our note. Any other comments or recommendations are welcome.

**Citation:**

Saxena RK. Estimation of canopy reservoir capacity and oxygen-18 fractionation in throughfall in a pine forest. Nord Hydrol 1986, 17:251–260.

---

## Author Comment (AC2)

Reply to the referee#2' comments:

We acknowledge the comments made by the Referee#2, that we will try to answer and take into account in order to improve our manuscript.

- The first reason we decided to publish this technical note was because we could not find any other published experimental validation of the Gonfiantini (1986) equation where evaporating waters experienced time periods when heavy isotopes were depleted instead of enriched.

Indeed, this was also an opportunity to demonstrate an example of the errors that can be incurred if the meteorological data are just time-averaged, as we did in Figures 4 and A2. However, these graphs show the errors in modelling the isotopy of the evaporating water, but not the errors on the inverse calculation of the volume of evaporated water from its isotopic composition, which is the most frequent target as stated in the introduction.

To this end, we applied an inversion of the equation (1):

$$(1 - x) = \left(\frac{\partial_L - \partial^*}{\partial_O - \partial^*}\right)^{(1/m)} \qquad \qquad \text{(R-1)}$$

Where $\delta_O$ represents the isotopy of the pool water at the beginning of the experiment, i.e. the 'original' water in real applications that can be obtained by the intersection of the LEL and the LMWL (e.g. Benettin et al., 2018), and $\delta_L$ represents the isotopy of the water obtained at every visit. The other variables and parameters are the same as for equation (1).

[Figure]

**Figure R- 3: Simulation of the relative volumes of evaporating waters applying equation (R-1) and comparing the use of both unweighted and ETP-weighted meteorological parameters. Gaps in the lines correspond to mathematical errors (non-real number results) of equation (R-1).**

The application of this equation to [18]O is shown in Figure R-3. When time-averaged (unweighted) meteorological data were used, the equation (R-1) predicted slightly smaller residual relative water volumes than observed for the 7 first observations, but it was in mathematical error for the remaining observations. These errors are due to the fact that the limiting water isotopy $\delta^*$ had values between the original $\delta_O$ and the terminal $\delta_L$ ones, an impossible arrangement because, following equation (1), evaporation will approximate the isotopy of water towards $\delta^*$ by increasing or decreasing its value, but it cannot modify the

isotopy of water by crossing the $\delta^*$ value. Figure 4a shows indeed that (unweighted) $\delta^*$ values were smaller than the observed $\delta_L$ values for the latter two-thirds of the experiment.

When the equation (R-1) was applied with ETP-weighted meteorological data, the results were much better for most of the observations, although some mathematical errors also occurred (line discontinuities). These errors do occur because the real $\delta_O$ of the evaporating process is not the value at the start of the experiment, but the $\delta_L$ value of the former step. In Figure 6a, these samples correspond to those that underwent $^{18}O$ depletion instead of enrichment: the corresponding observed $\delta$ values where smaller than the preceding ones but larger than the corresponding limiting $\delta^*$ values, so the application of equation (R-1) yields adequate results at the weekly step scale but not at the full experiment scale. The analysis at the weekly step scale are not feasible when time-averaged (unweighted) meteorological data were used, because the limiting $\delta^*$ values were strongly underestimated.

In fact, Figures 2a and R-3 demonstrate the limitations of the isotopic method for assessing the residual volume of water in a pool. This method would require a monotonic change in the isotopy for a decreasing volume of water, but this requirement fails when the isotopy of the pool water comes close to the steady ratio $\delta^*$. When this occurs, evaporation may continue with either sustained or decreasing concentrations of heavy isotopes.

- Although we have 5-minute weather data, our data on water level and rain and pool water isotopy were obtained at the weekly step. Therefore, although we recognise its possible interest, we discarded doing any simulation with a smaller time step, because we could not validate the results, losing the main purposes of our work. We acknowledge the suggestion and retain it for further investigations on the subject.

*L28 the cited papers did not all use weekly-monthly means.*

- We are amending this mistake in the revised version

*L35 bidirectional exchange is ubiquitous, not just when humidity is high.*

- We will modify this sentence following the writing of the cited authors and taking into account this recommendation.

*L45 isotope equilibration field studies have been conducted across a range of climates. The novelty of a subhumid climate is not great; e.g., the cited works by Gonfiantini include field data from Italy.*

- Yes. This is right, we are changing the sentence adequately.

*L67 it is a fine distinction, but the \*expected\* isotopic composition was modeled as Eq 1.*

 - Modeled instead of calculated is being stated

*L69 Eq 1 is explicitly derived "assuming that the evaporation conditions remain unchanged" (Gonfiantini 1986, eq 7), so it is no surprise that it does not perform well at weekly timescales.*

- The results show that the equation works reasonably well at weekly timescales when parameterized with ETP-weighted air temperature and relative humidity, as shown in Figures 6 and A4.

*Fig 3 d-precipitation does not appear in Eq 2; I assume this should be d-A?*

- In equation (2), $\delta^*$ are indeed calculated from $\delta_A$ (atmospheric moisture), which is derived from $\delta_P$ (precipitation) using equation (4). The value of both $\delta_P$ were set at =0 for designing the graph in order to obtain both $\delta^*$ =0 when air is at saturation.

*L100, L124-126, L129, L132 text duplicates figure captions with no additional information.*

- More explanations will be added to the text.

*L105 apply how? weekly means?*

- At any time step. This was written in order to show that equation (1) can simulate both depletion and enrichment. We can change the sentence into: "In fact, when equation (1) is applied to changes in pan water volume, the isotopic composition of the residual water ($\delta_L$) can approach $\delta^*$ either following trends of both enrichment and depletion of heavy isotopes".

*L106 but there is a 4-week period when d18O was increasing in the pan while d\* was less than the pan.*

-This behaviour is shown in Figure 4a, where $\delta^*$ values were underestimated because time-averaged temperature and humidity were used, but not in Figure (6a) obtained with ETP-weighted parameters. We will warn in the caption of the Figure (4) that these results are inadequate.

*L108 because d\* is completely theoretical and not a measured quantity, it is not clear why something "might" cause a decrease in d\*. Why is there any question? Similarly, L109-110 is simply restating the theory being applied, with no original content being contributed by the experiment.*

[Figure]

**Figure R 1: weekly estimates of $^{18}$O concentrations; steady ($\delta^*$) and in air moisture ($\delta a$), in relation to relative humidity. Vertical scales are offset 30 $^0/_{00}$ from each other.**

This refers to a time sequence: for the same isotopy of the atmospheric moisture, a temporal increase in relative humidity will determine (equation 2) a decrease in $\delta^*$, as shown in Figure 3, and also in the enclosed figure R1. This explains most of the temporal variation in $\delta^*$ shown in Figure 6a. We can delete the citation to Craig et al. (1963), but the original outcome of the experiment is to show (Figure 6a) that the water in the pan was depleted in heavy isotopes during several fragmented weeks during evaporation, and that these depletion events were adequately simulated by the Gonfiantini (1986) equation because an increase in air relative humidity determined a decrease of the $\delta^*$ value below the antecedent $\delta_O$ value in the pan

water. In other words, we observed and simulated using well known methods that water evaporation may cause weekly events of depletion in heavy isotopes of the evaporating water without the need of a high salinity of the water that would induce its progressive depletion instead of an irregular one.

*L111 there are no methods presented that would allow these mass balance estimates. Was the mass or volume of water in the pan measured each time? If so, please consider presenting those data instead of the calculated 16O mass. L114 suggests volume data are available.*

- Yes, we missed to explain in the methods section that the water volume was measured at every weekly visit, as shown in figures 4a and 6a.

We deemed that it was not necessary to describe how the amounts of light isotopes were calculated for the balance, but this was made as it follows:

First, the mass of water in moles $M_w$ was obtained for every visit from its volume using a density of 0.9976 kilograms per litre and a molar mass of 18.015 grams per mole. Small changes in these values due to the variation in heavy isotope concentrations were not taken into account because they are mutually cancelled out.

Then, $R_{sa}$ sample isotope ratios where obtained for each of the two heavy isotopes from the corresponding δ values:

$$R_{sa} = (\partial/1000 + 1) \cdot R_{st} \tag{R2}$$

Where $R_{st}$ are the isotopic ratios of the VSMOW standards which were taken as 1/6420 for $^2$H and 1/498.7 for $^{18}$O.

Finally, the mass in moles of the light isotopes $M_l$ were obtained for every sample and isotope using:

$$M_l = M_w \cdot n/(1 + R_{sa}) \tag{R3}$$

Where *n* is the number of atoms in each water molecule: 1 for oxygen and 2 for hydrogen.

*L118 I suggest not using "RH" because "h" is already defined as the same thing.*

-Yes, we will use "*h*" for relative humidity.

*L124 "d*18O" is not a concentration, it is a deviation.*

-Yes, we will use only the term 'value' for the denomination of δ, as done in most publications

*Fig A2 the meaning of the solid lines is not specified.*

- These are respectively a linear and a second order polynomial without any modelling purpose that are shown only as visual references. The equations will be shown in the final graphs.

*L125, Fig 5, Fig A2 details of the methods to estimate PET are needed.*

- We will state in the figure captions that PET was estimated using the Allen (1998) method. The estimation of PET were made as operationally recommended by the FAO in Allen et al (1998); we did not describe it the methods section because we did not introduce any

modification in the method, this is a well-known procedure and its description would need a much longer extension of the technical note.

*L136 relevant to what?*

- Relevant to the difference between isotope fractionation by a distillation process and by evaporation in a natural environment. We will try to improve the clarity of the sentence.

*L138-140 I do not understand the point being made about rainfall and humidity and d*. It appears the sentence assumes something about the relationship between rainfall and isotopic composition of atmospheric water vapor, but their relationship is irrelevant to d* and only the vapor matters. It is of no importance to this statement that the isotopic composition of rainfall was used as a surrogate for the isotopic composition of vapor in this experiment.*

- There are several publications that claim that the isotopic composition of the air moisture is very relevant to the fractionation of evaporating waters, but equation (2), used to design the standard Figure 3 and the above Figure R1 showing experiment outcomes demonstrate that this is only true when relative humidity is high, but the values of $\delta^*$ rapidly increase when relative humidity decreases, to an extent that the variations of atmospheric moisture isotopy become marginal (may explain up to 8% of the $\delta^*$ variance in Figure R1).

We will try to explain this better.

*L142 what is a "heavy isotope depletion period"? It is not clear which of the three nouns are being modified by "heavy." It is also not clear what a "depletion period" is L143. Are these referring to periods when d18O in the pan become more negative?"*

- We wanted to state "the susceptibility of the occurrence of periods in which water evaporation causes isotope depletion instead of enrichment must be taken into account."
Yes, these refer to the samples (weeks) when the $\delta$ values decrease instead of increase respect to the preceding ones. We will modify these final remarks using the results corresponding to the new Figure R3 above.

---

## Author Response (AR1)

**Referee#1:**

We thank the reviewer for his comments, which show us some aspects of our note that we did not explain well enough.

- The intended more original aspect of the note is not the role of diurnal variations, but the evidence of periods of heavy isotope depletion during water evaporation, along with the adequate prediction of these periods using customary fractionation equations parameterized with air conditions.

In the recent literature, heavy isotope depletion of water is usually attributed to exchange with atmospheric moisture in equilibrium conditions, whereas our results show that it can happen during evaporation when relative humidity is still far from saturation. The early definitions of $\delta*$ concentrations are not only the "isotopic composition which a water reaches in its final evaporation stages", but also "If h> 50% the isotopic composition rapidly approaches the limiting value A/B ($\delta*$) and then remains practically constant, simulating the occurrence of an isotopic steady state in the final evaporation stage" (Gonfiantini, 1986).

Indeed, Craig et al. (1963) already stated that "The deuterium and oxygen 18 concentrations of water evaporating into air of nonzero humidity do not follow the simple batch distillation equation but increase asymptotically to a stationary isotopic state as the mass of water decreases to zero".

These concepts are very relevant, because If the $\delta*$ concentrations are understood as only reached in the "final evaporation stages" (as reiterated in several recent articles) there is no way to understand how evaporation can result in depletion of heavy isotopes. Conversely, if these $\delta*$ concentrations are understood as "a stationary isotopic state" that is rapidly approached when the relative humidity of the air is rather high, It is easy to understand that changes in atmospheric conditions can induce the decrease of $\delta*$ concentrations below the concentrations already reached in the pan water, and determine the depletion of heavy isotopes during evaporation.

Several authors recommended the use of evaporation flux-weighted atmospheric conditions for isotopic fractionation studies, but we found no evaluation of the errors that can be incurred if these recommendations are not followed.

-The sentence "rapid molecular exchange of isotopes between the water body and the atmospheric vapour, which predominates over the net isotopic effect of a simple evaporation process" is not ours, but quoted from Craig et al., 1963. We acknowledge the proposal of more updated citations, but we try to show that our findings can be explained by the early developments made many years ago on the isotopy of evaporating open water bodies.

-We did not explain well ourselves with our sentence "In drier conditions, these $\delta*$ rapidly increase with decreasing air humidity and become detached from precipitation and atmospheric moisture isotopic content". We wanted to remark that, following eq. (2) and the resulting Fig (3), when the air is dry, the $\delta*$ values sharply increase to an

extent that the variations of atmospheric moisture isotopy become poorly relevant. Indeed, the results in our experiment show that the observed temporal changes in air relative humidity become much more important than those in air moisture isotopy, as estimated from precipitation isotopy. The new figure 6 shows the dependence of weekly $\delta*^{18}O$ and air moisture $\delta^{18}O$ estimates on relative humidity along the experiment. Air relative humidity explains 92% of $\delta*$ $^{18}O$ variance while moisture isotopy explains only 25%.

[Figure]

**Figure 1: Weekly estimates of $^{18}O$ isotopic composition, limiting ($\delta*$) and in air moisture ($\delta_A$), in relation to relative humidity. Vertical scales are offset 30 $^0/_{00}$ from each other.**

- We thought that showing the temporal variation of $\delta*$ values was sufficient to explain why we have decreasing isotopic composition at the late intervals. The comment leads us to add a new figure 7, following the one proposed by A. Rhode (cited by Saxena, 1986), which can help to understand the behaviour of our pan water.

[Figure]

Figure 2: Evolution of the weekly differences between the isotopy of water and air, plotted in relation to air relative humidity (from bottom to top). The green line that connects the $\delta^*$-$\delta_A$ points is a second order polynomial, shown only as visual reference.

In this figure, the differences in $\delta^{18}O$ between water and air moisture are plotted against air relative humidity, and the differences respect to $\delta^*$ values split the plot into a lower part where the differences in water isotopy are lower than that of $\delta^*$ and an upper part where these are higher. According to equation (1), water conditions located on the $\delta^*$ curve undergo evaporation without any isotopic change (steady state or dynamic equilibrium), while evaporating waters located outside this curve tend to move towards it, becoming enriched (upward) or depleted (downward). Minor vertical displacements of the points may be due to changes in the isotopy of the air moisture.

The isotopically depleted initial conditions (lower points) of the pan water determined its progressive enrichment during evaporation in spite of wide changes in air relative humidity. But by the ninth week of the experiment, pan water was already so enriched that a relevant increase in air relative humidity caused a strong decrease of $\delta^*$ and moved the situation of the pan water point to the 'depletion' area of the graph. Since this event, successive alternations in relative humidity determined isotopic enrichment and depletion periods, as already shown in Figures 2 and 6. We hope that these two new figures will help to show the changing environmental conditions along the experiment.

- We used air relative humidity because this is requested in the equations we used. Before using reference evapotranspiration for weighting relative humidity we tested several simpler options such as global radiation, vapour pressure deficit, as well as the combination of mean, minimum and maximum daily relative humidity. We agree that Penman formulation would be more adequate than Penman-Monteith reference formulation, but parameterisation of Penman-Monteith equation to 5-minute steps is

more physically sound and, although we did not use it for estimating pan evaporation, it predicted the decrease of its level with a determination coefficient of $r^2$=0.994.

- Pan level and the isotopy of both pan and precipitation waters were observed at weekly intervals, so it was necessary to aggregate air temperature and relative humidity to these periods for applying equations. The values of $\delta^*$ were obtained for every weekly step using equation (2). Pan evaporation $x$ was obtained from observed water level changes in the pan and the simulated isotopy $\delta_L$ of pan water was calculated using equation (1). For the first step, the original isotopy of the pan water was used as the initial isotopy $\delta_0$, and for the subsequent time steps, the isotopy $\delta_L$ obtained in the former step was used as the initial isotopy $\delta_0$ for the new step.

Citation:

Saxena RK. Estimation of canopy reservoir capacity and oxygen-18 fractionation in throughfall in a pine forest. Nord Hydrol 1986, 17:251–260.

**Referee #2:**

We acknowledge the comments made by the Referee#2, that we took into account in order to improve our manuscript.

- The first reason we decided to publish this technical note was because we could not find any other published experimental validation of the Gonfiantini (1986) equation where evaporating waters experienced time periods when heavy isotopes were depleted instead of enriched.

Indeed, this was also an opportunity to demonstrate an example of the errors that can be incurred if the meteorological data are just time-averaged, as we did in Figures 4 and A2. However, these graphs show the errors in modelling the isotopy of the evaporating water, but not the errors on the inverse calculation of the volume of evaporated water from its isotopic composition, which is the most frequent target as stated in the introduction.

To this end, we applied an inversion of the equation (1):

$$(1-x) = \left(\frac{\partial_L - \partial^*}{\partial_O - \partial^*}\right)^{(1/m)} \tag{11}$$

Where $\delta_O$ represents the isotopy of the pool water at the beginning of the experiment, i.e. the 'original' water in real applications that can be obtained by the intersection of the LEL and the LMWL (e.g. Benettin et al., 2018), and $\delta_L$ represents the isotopy of the water obtained at every visit. The other variables and parameters are the same as for equation (1).

[Figure]

**Figure 3: Simulated versus observed residual volumes of evaporating water applying equation (11) and comparing the use of both unweighted and PET-weighted meteorological parameters. Gaps in the lines correspond to lacking points due to mathematical errors (non-real number results) of equation (11).**

The application of this equation to [18]O is shown in the new figure (11). When time-averaged (unweighted) meteorological data were used, the equation (R-1) predicted slightly smaller residual relative water volumes than observed for the 7 first observations, but it was in mathematical error for the remaining observations. These errors are due to the fact that the limiting water isotopy $\delta^*$ had values between the original $\delta_O$ and the terminal $\delta_L$ ones, an impossible arrangement because, following equation (1), evaporation will approximate the

isotopy of water towards $\delta*$ by increasing or decreasing its value, but it cannot modify the isotopy of water by crossing the $\delta*$ value. Figure 4a shows indeed that (unweighted) $\delta*$ values were smaller than the observed $\delta_L$ values for the latter two-thirds of the experiment.

When the equation (11) was applied with ETP-weighted meteorological data, the results were much better for most of the observations, although some mathematical errors also occurred (line discontinuities). These errors do occur because the real $\delta_O$ of the evaporating process is not the value at the start of the experiment, but the $\delta_L$ value of the former step. In Figure 6a, these samples correspond to those that underwent $^{18}O$ depletion instead of enrichment: the corresponding observed $\delta$ values where smaller than the preceding ones but larger than the corresponding limiting $\delta*$ values, so the application of equation (R-1) yields adequate results at the weekly step scale but not at the full experiment scale. The analysis at the weekly step scale are not feasible when time-averaged (unweighted) meteorological data were used, because the limiting $\delta*$ values were strongly underestimated.

In fact, Figures 2a and 8 demonstrate the limitations of the isotopic method for assessing the residual volume of water in a pool. This method would require a monotonic change in the isotopy for a decreasing volume of water, but this requirement fails when the isotopy of the pool water comes close to the steady ratio $\delta*$. When this occurs, evaporation may continue with either sustained or decreasing concentrations of heavy isotopes.

- Although we have 5-minute weather data, our data on water level and rain and pool water isotopy were obtained at the weekly step. Therefore, although we recognise its possible interest, we discarded doing any simulation with a smaller time step, because we could not validate the results, losing the main purposes of our work. We acknowledge the suggestion and retain it for further investigations on the subject.

*L28 the cited papers did not all use weekly-monthly means.*

- We amended this mistake in the revised version

*L35 bidirectional exchange is ubiquitous, not just when humidity is high.*

- We modify this sentence into: "Isotopic exchange that induces the depletion rather than enrichment in heavy isotopes of the evaporating water has been identified..."

*L45 isotope equilibration field studies have been conducted across a range of climates. The novelty of a subhumid climate is not great; e.g., the cited works by Gonfiantini include field data from Italy.*

- Yes. This is right, but not to complete dryness; we changed the sentence into: "an artificial pan was set up and subjected to evaporation to complete dryness in a location with a sub-humid climate, as a counterweight to the more frequent studies in dry climates".

*L67 it is a fine distinction, but the \*expected\* isotopic composition was modeled as Eq 1.*

- Modeled instead of calculated is now stated

*L69 Eq 1 is explicitly derived "assuming that the evaporation conditions remain unchanged" (Gonfiantini 1986, eq 7), so it is no surprise that it does not perform well at weekly timescales.*

- The results show that the equation works reasonably well at weekly timescales when parameterized with ETP-weighted air temperature and relative humidity, as shown in Figures 5 and A5.

*Fig 3 d-precipitation does not appear in Eq 2; I assume this should be d-A?*

- In equation (2), $\delta*$ are indeed calculated from $\delta_A$ (atmospheric moisture), which is derived from $\delta_P$ (precipitation) using equation (4). The value of both $\delta_P$ were set at =0 for designing the graph in order to obtain both $\delta*$ =0 when air is at saturation.

*L100, L124-126, L129, L132 text duplicates figure captions with no additional information.*

- More explanations are now added to the text.

*L105 apply how? weekly means?*

- At any time step. This was written in order to show that equation (1) can simulate both depletion and enrichment. This sentence has been deleted and the fact that equation (1) may predict both enrichment and depletion is stated when the equation is explained: "It is worth emphasizing that this equation does not determine that evaporation induces an increase of $\delta_L$ with respect to $\delta_0$ (enrichment) but that $\delta_L$ approaches $\delta*$ in either direction (enrichment or depletion)."

*L106 but there is a 4-week period when d18O was increasing in the pan while d* was less than the pan.*

-This behaviour is shown in Figure 2a, where $\delta*$ values were underestimated because time-averaged temperature and humidity were used, but not in Figure (5a) obtained with ETP-weighted parameters. We warn now in the caption of the Figure (5) that these results are inadequate.

*L108 because d* is completely theoretical and not a measured quantity, it is not clear why something "might" cause a decrease in d*. Why is there any question? Similarly, L109-110 is simply restating the theory being applied, with no original content being contributed by the experiment.*

This refers to a time sequence: for the same isotopy of the atmospheric moisture, a temporal increase in relative humidity will determine (equation 2) a decrease in $\delta*$, as shown in Figure 3, and also in the enclosed figures 6 and 7. This explains most of the temporal variation in $\delta*$ shown in Figure 5a. We could delete the citation to Craig et al. (1963), but the original outcome of the experiment is to show (Figure 5a) that the water in the pan was depleted in heavy isotopes during several fragmented weeks during evaporation, and that these depletion events were adequately simulated by the Gonfiantini (1986) equation because an increase in air relative humidity determined a decrease of the $\delta*$ value below the antecedent $\delta_0$ value in the pan water. In other words, we observed and simulated using well known methods that water evaporation may cause weekly events of depletion in heavy isotopes of the evaporating water without the need of a high salinity of the water that would induce its progressive depletion instead of an irregular one.

[Figure]

**Figure 4: Weekly estimates of $^{18}$O isotopic composition, limiting ($\delta^*$) and in air moisture ($\delta_A$), in relation to relative humidity. Vertical scales are offset 30 $^0/_{00}$ from each other.**

[Figure]

**Figure 5: Evolution of the weekly differences between the isotopy of water and air, plotted in relation to air relative humidity (from bottom to top). The green line that connects the $\delta^*$-$\delta_A$ points is a second order polynomial, shown only as visual reference.**

*L111 there are no methods presented that would allow these mass balance estimates. Was the mass or volume of water in the pan measured each time? If so, please consider presenting those data instead of the calculated 16O mass. L114 suggests volume data are available.*

- Yes, we missed to explain in the methods section that the water volume was measured at every weekly visit, as shown in figures 4a and 6a.

We deemed that it was not necessary to describe how the amounts of light isotopes were calculated for the balance, but this was made as it follows:

First, the mass of water in moles $M_w$ was obtained for every visit from its volume using a density of 0.9976 kilograms per litre and a molar mass of 18.015 grams per mole. Small changes in these values due to the variation in heavy isotope concentrations were not taken into account because they are mutually cancelled out.

Then, $R_{sa}$ sample isotope ratios where obtained for each of the two heavy isotopes from the corresponding δ values:

$$R_{sa} = (\partial/1000 + 1) \cdot R_{st} \tag{9}$$

Where $R_{st}$ are the isotopic ratios of the VSMOW standards which were taken as 1/6420 for $^2$H and 1/498.7 for $^{18}$O.

Finally, the mass in moles of the light isotopes $M_l$ were obtained for every sample and isotope using:

$$M_l = M_w \cdot na/(1 + R_{sa}) \tag{10}$$

Where *na* is the number of atoms in each water molecule: 1 for oxygen and 2 for hydrogen.

*L118 I suggest not using "RH" because "h" is already defined as the same thing.*

-Yes, we now use "*h*" for relative humidity.

*L124 "d*18O" is not a concentration, it is a deviation.*

-Yes, we now use only the term 'value' for the denomination of δ, as done in most publications

*Fig A2 the meaning of the solid lines is not specified.*

- These are respectively a linear and a second order polynomial without any modelling purpose that are shown only as visual references. The equations are now shown in the final graphs.

*L125, Fig 5, Fig A2 details of the methods to estimate PET are needed.*

- We state in the methods that PET was estimated using the Allen (1998) method. The estimation of PET were made as operationally recommended by the FAO in Allen et al (1998); we did not describe it the methods section because we did not introduce any modification in the method, this is a well-known procedure and its description would need a much longer extension of the technical note.

*L136 relevant to what?*

- Relevant to the difference between isotope fractionation by a distillation process and by evaporation in a natural environment. We will try to improve the clarity of the sentence.

*L138-140 I do not understand the point being made about rainfall and humidity and d*. It appears the sentence assumes something about the relationship between rainfall and isotopic composition of atmospheric water vapor, but their relationship is irrelevant to d* and only the vapor matters. It is of no importance to this statement that the isotopic composition of rainfall was used as a surrogate for the isotopic composition of vapor in this experiment.*

- There are several publications that claim that the isotopic composition of the air moisture is very relevant to the fractionation of evaporating waters, but equation (2), used to design the standard Figure 3 and the above Figure 6 showing experiment outcomes demonstrate that this is only true when relative humidity is high, but the values of δ* rapidly increase when relative humidity decreases, to an extent that the variations of atmospheric moisture or rainfall isotopies become marginal.

*L142 what is a "heavy isotope depletion period"? It is not clear which of the three nouns are being modified by "heavy." It is also not clear what a "depletion period" is L143. Are these referring to periods when d18O in the pan become more negative?"*

- We deleted this sentence and wrote the following one: " Evaporation of water does not always induce heavy isotope enrichment, but may progress without isotopic change in a steady state process when the composition of evaporating water is equal to the limiting δ* value, or it can lead to isotopic depletion when it exceeds this value.

In this experiment, we observed several alternating weeks of heavy isotope enrichment and depletion during evaporation of pan water. These events were successfully simulated using classical equations and attributed to temporal increases of air relative humidity and corresponding decreases of the limiting δ* values, below the composition reached by the evaporating water."

**List of relevant changes**

Many changes have been made to the wording of the manuscript according to the referees' comments, without changing its structure or meaning.

Three new equations (9, 10 and 11) were added.

The old Figure 1 was deleted.

Four new figures (6, 7, 8 and A4) were added

---

## Author Response (AR2)

**Reviewer's comments:**

There is substantially more content included in this revision, and most of it is helpful. However, I became lost in the organization; much of the new material is new analyses that are explained in the results rather than the methods. Finally, now that the flux weighting is easier to understand, it brought to mind two questions: (1) should the PET weighting be modified by a pan coefficient, or does that cancel out? And (2) what are the consequences for using a mass-based model of isotopic evolution rather than a time-based model, i.e., what are the consequences of ignoring isotopic equilibration with atmospheric vapor at times when evaporation is small? Non-evaporative equilibration should be a greater effect when pan volumes are low and pan-water isotopic composition is far from the atmosphere, both of which are most important later in the experimental period. Reliance on the mass-based model has resulted in an error of interpretation that evaporation can cause isotopic depletion, when in fact it is isotopic exchange that is responsible and that overwhelms the evaporative distillation (see common L214).

English suggestions: L46 "composition"; L84 try "require" instead of determine; L200 "is" not feasible; L204 fails "if" … and then omit comma L205.

L60 this last sentence is ambiguous: errors occur when both conditions are the case, or is only one of them enough?

L109 both light- and heavy species certainly did both evaporate and condense because exchange with vapor is ongoing all the time. I think the intent here is to investigate net mass fluxes?

L120 evaporation is not zero at night. More important, however, is that equilibration between liquid and vapor does not stop at night. Gonfiantini's (1986) equation describes progression toward equilibrium through mass-loss space as evaporation progresses in steady-state conditions. The experiment, therefore, at least partially tests whether there were important variations in atmospheric vapor at night that affected the assumption of equilibration through mass space.

L120 a pan coefficient is appropriate here as well.

L131 LEL and LMWL are not defined.

L144 which one is the last figure? There is a caption that says everything that L144 does, and L144 can be removed and the figure cited on L145. This style of repeating caption material in the text is consistent throughout sec 3, and I think it makes it difficult to follow the story.

L210 atmospheric vapor is not always in equilibrium with precipitation, even when humidity is high, so this sentence assumes too much. For example, humidity in coastal regions or wetlands may occasionally bear little resemblance to precipitation.

L214 importantly, evaporation does not cause depletion in this case. Rather, back-equilibration overwhelms the evaporative distillation. Just because the models are expressed in mass space does not mean that net mass flux is alone responsible for the fractionation.

L225 it is a little misleading to say that time averaging made a larger difference when residual volume was lower. It's true, but it directs attention to the wrong place. Time averaging introduces errors into alpha and h, but those do not depend on the residual volume. I think it's better to say that errors are propagated more strongly when volumes are low.

**Responses:**

We want to acknowledge the Editor for these notes that helped us to deliberate some aspects and improve the clarity of the paper.

- New analyses in the Results and Discussion section:

We reviewed the Results and Discussion section and did not found methods not described in the Data and Methods section. The main new material is Figure 7 that synthesises the isotopic evolution of the pool water during the experiment. This graph is just a new representation of the results of pool water analyses and equations (2) and (4) with some simple recalculations. Therefore, we prefer to present Figure 7 in the section it is currently in.

- Should the PET weighting be modified by a pan coefficient, or does that cancel out?:

As the final target of this work is the application of the method to a wide range of natural pools and the use of PET is not to predict pan water evaporation but only to weight the atmospheric conditions, we decided to test the operational FAO Penman-Monteith equation parameterized as for reference evapotranspiration, without any pan coefficient. When we compared the PET predicted *versus* observed decreases of water volume in the pan, we found that the pan coefficient decreased from an initial value of 0.80 to a final value of 0.42. We attributed this decrease to the increasing role of pan walls hindering the aeration of the water surface. Fortunately, as suggested by the reviewer, the pan coefficient cancels out when the resulting evaporative demand is used for weighting relative air humidity and temperature.

-What are the consequences for using a mass-based model of isotopic evolution rather than a time-based model, i.e., what are the consequences of ignoring isotopic equilibration with atmospheric vapor at times when evaporation is small?:

Our data consist of weekly measures of pan water volume along with isotopic composition of pan water and precipitation. With these data we deemed that the best option was using the Gonfiantini (1986) equation, , assuming that the atmospheric conditions and air moisture isotopic composition remain fairly constant between sampling dates. We have no temporally finer data on water volume and isotopic compositions adequate for a time-based modelling.

We first tested the hypothesis that fractionation can be predicted by evaporation following Gonfiantini (1986) equation parameterized with time-averaged h and T). This hypothesis was rejected due to the prediction of an earlier and deeper isotopic depletion than observed. Then we tested the hypothesis that fractionation can be predicted by evaporation following Gonfiantini (1986) equation parameterized with flux-averaged h and T. This approximation give us adequate results.

We do not deny some possible role of non-evaporative isotopic equilibration during the night but the experiment didn't allow us to validate it, this would require another experimental design.

- English suggestions: L46 "composition"; L84 try "require" instead of determine; L200 "is" not feasible; L204 fails "if" … and then omit comma L205: taken into account.

- L60 this last sentence is ambiguous: errors occur when both conditions are the case, or is only one of them enough?: this sentence has been rewritten accordingly.

- L109 both light- and heavy species certainly did both evaporate and condense because exchange with vapor is ongoing all the time. I think the intent here is to investigate net mass fluxes?: this sentence has been rewritten accordingly.

- L120 evaporation is not zero at night. More important, however, is that equilibration between liquid and vapor does not stop at night. Gonfiantini's (1986) equation describes progression toward equilibrium through mass-loss space as evaporation progresses in steadystate conditions. The experiment, therefore, at least partially tests whether there were important variations in atmospheric vapor at night that affected the assumption of equilibration through mass space: This would need data and discussion that is beyond the frame of this technical note. The purpose here is just to flux-weight the meteorological variables as recommended by several authors. We changed 'when evaporation is inactive' into 'when evaporation is the lowest'.

- L120 a pan coefficient is appropriate here as well: the fact that a pan coefficient was not necessary because it becomes cancelled out is stated here.

- L131 LEL and LMWL are not defined: LEL and LMWL are now defined here.

- L144 which one is the last figure? There is a caption that says everything that L144 does, and L144 can be removed and the figure cited on L145. This style of repeating caption material in the text is consistent throughout sec 3, and I think it makes it difficult to follow the story: This suggestion has been taken into account and several changes in the writing are made for avoiding repetition with figure captions.

- L210 atmospheric vapor is not always in equilibrium with precipitation, even when humidity is high, so this sentence assumes too much. For example, humidity in coastal regions or wetlands may occasionally bear little resemblance to precipitation: this sentence has been rewritten accordingly.

- L214 importantly, evaporation does not cause depletion in this case. Rather, back-equilibration overwhelms the evaporative distillation. Just because the models are expressed in mass space does not mean that net mass flux is alone responsible for the fractionation : this sentence has been rewritten accordingly.

- L225 it is a little misleading to say that time averaging made a larger difference when residual volume was lower. It's true, but it directs attention to the wrong place. Time averaging introduces errors into alpha and h, but those do not depend on the residual volume. I think it's better to say that errors are propagated more strongly when volumes are low: The role of both the cumulative effect of underestimation of $\delta^*$ and increased error propagation is now considered as causes of increasing errors in the later steps of the experiment.